# Quality of Life and Mental Health Status in Recovered COVID-19 Subjects at Two Years after Infection in Taizhou, China: A Longitudinal Cohort Study

**DOI:** 10.3390/brainsci12070939

**Published:** 2022-07-18

**Authors:** Juan Pan, Kai Zhou, Jing Wang, Yufen Zheng, Die Yu, Haixin Kang, Yanjie Zhang, Shuotao Peng, Tao-Hsin Tung, Bo Shen

**Affiliations:** 1Department of Clinical Laboratory, Taizhou Hospital, Zhejiang University, Linhai 317000, China; xs_panj@enzemed.com; 2Department of Clinical Laboratory, Taizhou Hospital of Zhejiang Province, Wenzhou Medical University, Linhai 317000, China; zhouk7018@enzemed.com (K.Z.); wjing@enzemed.com (J.W.); zhengyf@enzemed.com (Y.Z.); xs_yud@enzemed.com (D.Y.); xs_kanghx@enzemed.com (H.K.); xs_zhangyj@enzemed.com (Y.Z.); xs_pengst@enzemed.com (S.P.); 3Evidence-Based Medicine Center, Taizhou Hospital of Zhejiang Province, Wenzhou Medical University, Linhai 317000, China

**Keywords:** GHQ-12, PTSD, quality of life, COVID-19, recovered subjects, multidisciplinary

## Abstract

According to previous studies, mental status in 1-year COVID-19 survivors might range from 6–43%. Longer-term psychological consequences in recovered COVID-19 subjects are unknown, so we analyzed longer-term quality of life and mental status in recovered COVID-19 subjects at 2 years after infection. Among 144 recovered COVID-19 subjects in the Taizhou region, 73 and 45 completed face-to-face follow-ups at the first year and second year after infection, respectively, with a 61.7% follow-up rate. The questionnaire, which was administered at both follow-ups, included questions about quality of life, psychological health, and post-traumatic stress disorder (PTSD). The Mann-Whitney U test was used to the differences of each scale between the first and second year. Among the 45 people who completed both follow-up visits, the incidence of psychological problems was 4.4% (2/45) in the first year, and no new psychological abnormalities were observed in the second year. Quality of life improved, while the General Health Questionnaire (GHQ-12) and Impact of Event Scale-Revised (IES-R) scores did not improve over time. The incidence of mental disorders was lower than those in previous studies. Multidisciplinary management for COVID-19 in this study hospital may have reduced the frequency to a certain extent. However, among those with mental health problems, such problems may exist for a long time, and long-term attention should be given to the psychological status of recovered COVID-19 subjects.

## 1. Introduction

The coronavirus disease 2019 (COVID-19) epidemic continues to spread more than two years after the initial outbreak of COVID-19. Many countries have experienced waves of outbreaks, and new cases both at home and abroad are being reported every day. With the emergence of new variants, additional waves are expected. Proliferation speed and infectivity have increased due to the emergence of the Omicron variant [1], and even among those with complete vaccination, it is possible to become infected with the virus, making the disease highly contagious and difficult to control. The United States and the United Kingdom have advocated measures to exist with the virus rather than eradicating and eliminating it [2]. China implemented lockdown, quarantine, observation, and nucleic acid testing measures to promote the “dynamic elimination in society” policy, which reduced the speed of transmission and the number of infected people [3,4]. However, the psychological burden among the public and recovered COVID-19 subjects may be increased.

Manifestations in patients with immune inflammation due to severe acute respiratory syndrome coronavirus 2 (SARS-CoV-2) infection range from mild illness to acute respiratory distress syndrome and even death. Proinflammatory factors may have certain impacts on mental state, inducing post-traumatic stress disorder (PTSD) or depression, and seriously affect physical health [5,6]. Studies have found that the incidence of anxiety and depression during hospitalization can reach 50% [7], and at two months after discharge, nearly 40% of recovered patients had a mental disease [8]. A meta-analysis suggested that recovered COVID-19 subjects had higher rates of anxiety, depression, and PTSD after discharge at 4 months and 6 months than other patients [9].

Compared with the number of studies conducted in the initial stage of the disease, there are few studies on long-term mental health, and the longest follow-up period is one year at present. Mario Gennaro Mazza et al. found that after 12 months, 45% of recovered subjects had at least one mental health problem, such as anxiety, depression, or PTSD [10]. A longitudinal cohort study found that the mental and cognitive impairment in COVID-19 survivors had not improved after 1 year compared to that at the 3-month follow-up, and neurological symptoms still existed [11]. Lixue Huang et al. found that the incidence rates of anxiety and depression at the 12-month follow-up were slightly higher than those at the 6-month follow-up [12]. In addition, previous severe acute respiratory syndrome (SARS) and Middle East respiratory syndrome (MERS) events were shown to cause serious disturbance and damage to survivors’ psychosocial function even after many years [13]. COVID-19-recovered subjects could have psychological problems for many years.

At present, there is little knowledge about changes in the long-term quality of life, mental health, and PTSD status in recovered COVID-19 subjects. It is very important for patients to pay attention to long-term mental and psychological problems, as anxiety, depression, and PTSD can have a certain impact on quality of life [14]. This study evaluated quality of life, the psychological status, and the PTSD rate in recovered COVID-19 subjects at one year and two years after discharge and observed whether quality of life, the psychological status, and PTSD rate improved with the extension of recovery time.

## 2. Materials and Methods

### 2.1. Study Design and Participants

From 17 January to 10 March 2020, 144 confirmed COVID-19 patients who were diagnosed by real-time reverse transcriptase-polymerase chain reaction (RT-PCR) from a nasopharyngeal and/or throat swab and chest CT according to the Pneumonia Diagnosis and Treatment Scheme for New Corona Virus Infections (Trial Edition 5, Revision) in Taizhou, Zhejiang, were recruited. The investigators performed telephone follow-up in 144 subjects, which lasted at least 6 min; in the follow-up, the investigators explained the purpose of the study, face-to-face follow-up time, and the confidentiality of the respondents’ information. They attended face-to-face follow-ups 1 year (9 February 2021 and 10 February 2021) and 2 years after discharge (9 February 2022 and 10 February 2022). Excluding those who refused to participate or were lost to follow-up (*n* = 42), who could not be contacted (*n* = 6), and who were living outside Taizhou (*n* = 23), 73 subjects completed the follow-up at 1 year, and 45 subjects completed both follow-ups as detailed in Figure 1. Each subject signed an informed consent form. The study was approved by the ethics committee of Taizhou Hospital in Zhejiang Province. This study was registered in China’s clinical trial registry with the registration number ChiCTR2100048440. All programs were carried out according to the standards of our ethics committee and adhered to the tenets of the Declaration of Helsinki.

### 2.2. Data Sources/Measurement

All eligible recovered COVID-19 subjects were interviewed in-person by trained doctors, and the follow-up questionnaires included two sections; the first section focused on demographic characteristics, including sex, age, body mass index (BMI), severity of COVID-19, comorbidities, and length of stay (LOS). The second section included assessments for dyspnea, quality of life, mental health status, and PTSD.

The modified Medical Research Council dyspnea scale (mMRC) was used to describe the severity of dyspnea, with higher scores indicating a greater degree of dyspnea [12]. Quality of life was measured by using the 5-level version of the EuroQol 5-dimensional questionnaire (EQ-5D-5L), which includes the five dimensions of mobility, self-care, daily activities, pain/discomfort, and anxiety/depression [15]. Each dimension is classified into five levels, and the final continuous outcome ranges from 0 to 1 [16]; the higher the score is, the better the subject’s physical health is. To measure the impact of COVID-19 on mental health, we used the General Health Questionnaire (GHQ-12) (Cronbach’s α: 0.82). There are a total of 12 questions; the first two answers of each question are scored as 0 points, and the last two answers are scored as 1 point. The total score ranges from 0–12, and a total score ≥ 3 points indicates the existence of mental health problems [17,18]. The scale includes items to assess anxiety/depression, social dysfunction, and loss of confidence [19]. Referring to the Impact of Event Scale-Revised (IES-R), we assessed PTSD with five items [20]. Answers ranged from “not at all” to “extremely”, with scores of 0–4, respectively, for an overall score ranging from 0–20 (α = 0.77); the higher the score is, the more severe the PTSD is.

### 2.3. Statistical Methods

Categorical variables are expressed as frequencies and percentages, and continuous variables are expressed as means and standard deviations. The chi-square or Fisher’s exact test was used to measure demographic characteristics. Differences in the mMRC, EQ-5D-5L, GHQ-12, and IES-R scores were detected by the Mann-Whitney U test. IBM SPSS 26 software (SPSS Inc., Chicago, IL, USA) was used for data analysis, and *p* < 0.05 was considered statistically significant. Origin 2019 (OriginLab Corporation, Northampton, MA, USA) and GraphPad Prism version 8.3.0 (GraphPad Software, San Diego, CA, USA) were used to construct the graphs.

## 3. Results

### 3.1. Demographic Characteristics of the 45 Recovered COVID-19 Subjects Who Completed Both Follow-Ups and the 28 Recovered COVID-19 Subjects Who Completed Only the First Follow-Up

In total, 144 recovered COVID-19 subjects were included in this study in Taizhou, Zhejiang Province. After applying the exclusion criteria, 73 eligible subjects completed the first-year follow-up, and among them, 45 completed the second-year follow-up, with a completion rate of 61.7% (45/73). Table 1 shows comparisons of demographic characteristics between the 45 recovered COVID-19 subjects who participated in both follow-up visits and the 28 patients who participated in only the first follow-up visit. The results revealed that the patients who participated in both follow-up visits were older and had a larger number of underlying diseases than those who participated in only the first follow-up visit.

### 3.2. Basic Information of the 45 Recovered COVID-19 Subjects Who Completed Both Follow-Ups

Table 2 shows the basic information of and the scores of each scale for the 45 subjects. There were 23 males and 22 females. The median age was 51 years (interquartile range (IQR) 42–56). In total, 24% of the subjects had severe disease, 44% had comorbidities, and 55.6% had a LOS > 21 days. In the first and second years of follow-up, the mean values of the EQ-5D-5L were 0.90 (SD 0.12) and 0.95 (SD 0.09), respectively. The EQ-5D-5L scores were significantly different (*p* = 0.018); the quality of life in the second year was higher than that in the first year. The mean values of mMRC, GHQ-12, and IES-R scores are shown in Table 2, with no significant different between the two years on these scales. At the first and second follow-up among the 45 recovered COVID-19 subjects, the proportion of GHQ-12 scores ≥ 3 were 4.4% and 2.2%, respectively. In the second year, one person had persistent psychological abnormalities, while no new psychological abnormalities were observed.

### 3.3. Changes in the mMRC, EQ-5D-5L, GHQ-12, and IES-R Scores in the 45 Recovered Subjects

The changes in mMRC, EQ-5D-5L, GHQ-12, and IES-R scores in the 45 recovered COVID-19 subjects from the first year to the second year are shown in the heatmap in Figure 2A. The IES-R scores of some recovered COVID-19 subjects increased in the second year, while those of others decreased. EQ-5D-5L scores were increased in most patients in the second year compared with those in the first year. Figure 2B further describes the change trends of the scores for the assessment of dyspnea, quality of life, mental health, and PTSD in the recovered patients over two years. The quality of life of most of the recovered subjects increased in the second year, while some had increased IES-R and GHQ-12 scores in the second year. 

### 3.4. Changes in the Scales among the 45 Recovered Subjects in Different Groups

Figure 3 shows the detailed mean values of the mMRC, EQ-5D-5L, GHQ-12, and PTSD scores in the first and second years in the 45 recovered subjects stratified by sex, age, BMI, underlying disease, severity, and LOS. It was observed that the quality of life in the second year was slightly better than that in the first year. Male (*p* = 0.045) and short length of stay (*p* = 0.034) were higher in scales of EQ-5D-5L in the second year. Subjects who were female, aged ≥ 50 years, had a BMI ≤ 25, had mild disease, had a LOS > 21 days, and had comorbidities had higher PTSD averages than their counterparts. In females, the mean PTSD score was higher in the second year than in the first year, but the opposite was true in males. Patients with comorbidities or mild disease had a higher PTSD score, and the two-year mean value showed a downward trend. In the second year, the GHQ-12 score in the recovered subjects with severe disease increased. In subjects with a LOS > 21 days, the PTSD and GHQ-12 scores in the second year were higher than those in the first year.

## 4. Discussion

At present, there is a lack of studies on longer-term quality of life and psychological outcomes in recovered COVID-19 subjects; therefore, we conducted a longitudinal follow-up cohort study in recovered COVID-19 subjects in Taizhou at 1 and 2 years after discharge from the hospital. To our knowledge, this study represents the longest follow-up study on the psychology and quality of life of COVID-19 survivors to date. This study found that recovered COVID-19 subjects had a relatively low incidence of psychological problems, but the psychological harm caused did not improve with the extension of life and time; such problems may last for more than one year.

In the Taizhou region, the incidence of mental health problems among recovered COVID-19 subjects was 4.4% in the first year, and there were no new psychological abnormalities in the second year. A previous study found that the incidences of anxiety and depression in the urban Chinese population were 6.0% and 5.3%, respectively [21,22], and the age-normalized incidences of depression and anxiety in rural Chinese adults were 5.41% and 4.94%, respectively [23]. Previous 1-year follow-up studies in recovered COVID-19 subjects found that the incidence of psychological problems (only anxiety) was as low as 10.4% [10,12,24,25,26].

And browsing Table 3, studies on the mental health in 1-year COVID-19 survivors from different countries are presented, and the highest incidences of anxiety and depression were, respectively, 41.5% and 42.6%, and the lowest incidences of anxiety and depression were, respectively, 8.5% and 6%. Mario Gennaro Mazza et al. [10] and Yumiao Zhao et al. [27] found more than 40% with psychopathology. The incidence of mental diseases in this study was lower than those in previous studies. 

Studies have found that the rigor and timeliness of the government’s response to COVID-19 affected the physical and mental health of the population. Strict measures implemented by the government to contain the spread of COVID-19 benefit the physical health of populations, and the early implementation of strict lockdown measures can reduce the prevalence of depressive symptoms [28]. Vinod et al. found correlations between anxiety and depression and delayed treatment [29]. Therefore, we know that timely treatment and effective epidemic control have important impacts on psychology. Our hospitals have actively responded to the outbreak of COVID-19 by implementing effective management and timely treatment measures. Prehospital management comprises the online establishment of prevention and control teams who make online diagnoses and establish treatment models (including consultation, screening, and psychological intervention). These teams also promote online publicity and education, with foci on prehospital screening, early detection, and early treatment [30]. In confirmed COVID-19 inpatients, multiple systems may be involved, requiring the implementation of a multidisciplinary treatment strategy. The emergency department provides prehospital emergency care and triage. Auxiliary departments (the clinical laboratory, radiology, and ultrasonography) perform screening examinations, make diagnoses, and identify and address comorbid disorders according to the affected organ systems in the patients. Specialized departments provide symptomatic treatment. The department of psychiatry and psychology provides psychological interventions, and nurses provide health education and psychological comfort to patients receiving treatments such as immunosuppressive therapy, antiviral therapy, psychological counseling, and other symptomatic treatment in a timely manner. Multidisciplinary and timely, effective treatment may reduce psychological harm to a certain extent. In addition, there was a higher proportion of males in these studies, and shown in the Table 3, it was found in Mario Gennaro Mazza’s research that male psychological symptoms gradually increase over time [10]. In the study of Yumiao Zhao, there were 45.8% severe patients [27], while only 24.0% were severe subjects in this study. Moreover, there may be differences in the incidence also due to small, single-center sample sizes; different countries; and scale use.

In addition, we found that the GHQ-12 and IES-R scores in recovered COVID-19 subjects did not change significantly over the two-year study period, and some subjects had increased IES-R scores in the second year, indicating that recovered COVID-19 subjects may experience long-term mental and psychological problems, similar to the results of previous studies on SARS. Previous studies have found that psychiatric complications in SARS-infected patients can last for more than 2 years after the outbreak [37]. Mak et al. found that PTSD is one of the most common long-term psychiatric diagnoses in SARS survivors; 25.6% (23/96) of the subjects suffered from PTSD at 30 months after discharge, with less than 50% of survivors recovering from PTSD, and the recovery rate was slow [37]. Xia Hong et al. followed 70 SARS survivors for up to 4 years. They found that 28 survivors were diagnosed with PTSD at an average of 53 days after discharge, and 23 were diagnosed with PTSD at 46 months after discharge. PTSD symptoms did not improve significantly over time [38]. SARS survivors can suffer from persistent mental health problems, including anxiety, depression, and trauma, for up to 12 years [39]. In contrast, Mazza et al. showed that PTSD symptoms in recovered COVID-19 subjects improved over time [40]. Mental health outcomes in COVID-19 survivors may vary slightly depending on the questionnaire used, method applied (online or face-to-face), and time of follow-up, but there is a possibility of long-term psychological problems in recovered COVID-19 subjects. The mechanism of long-term mental and psychological problems in recovered COVID-19 patients is not clear; problems may be due to infection of the central nervous system by viruses or an abnormal immune responses to the virus [13,40,41]. Poyraz et al. pointed out that the risk of PTSD among COVID-19 survivors with long-term symptoms was 5 times higher than that in those without long-term symptoms [42]. In addition, the study found that 16.2% of the public believed that SARS patients could possibly transmit the virus 18 months after infection, and some groups discriminated against SARS patients [43], adversely affecting the positive help-seeking behavior and mental psychology of SARS patients [44]. According to a survey on stigmatization among COVID-19-infected persons, most patients felt stigmatized and discriminated against, and social discrimination and long-term symptoms may contribute to the existence of long-term mental and mental health conditions [45].

As of 5 April 2022, there were 15,355 new asymptomatic cases and 25,060 confirmed domestic cases (including 63 severe cases); however, the diagnosed patients had mostly asymptomatic or mild disease [46]. While it is possible to achieve cure through treatment, the psychological impact cannot be ignored, and we need to provide early psychological counseling and long-term psychological support and interventions. According to the mental health results of the subgroup analyses, women and patients with a longer LOS had a lower quality of life. Those were also the influential factors for the psychology of COVID-19 survivors [47,48], according to previous studies. Those kinds of subjects should be more cognizant of physical and mental health problems. It was found that women, older individuals, patients with a longer LOS, and patients with comorbidities should be cognizant of mental health problems. At present, the epidemic is not completely controlled, and newly diagnosed cases are still being reported. Long-term mental illness not only affects the quality of life and physical health of survivors but also may impose significant economic and social burdens in the post-pandemic world [49]. We suggest that early psychological intervention should be advocated.

COVID-19 has caused both physical and psychological harm in infected persons. Although the symptoms of COVID-19 caused by the highly infectious Omicron variant are milder than those caused by other variants, infection may still cause fear and impose a psychological burden. In addition to providing timely symptomatic treatment, medical institutions also need to encourage psychologists and psychiatrists to provide psychological counseling at the early stage and continuous psychological support in the later stage. A multidisciplinary treatment model may be more conducive to improvement of the physical and mental health of infected individuals.

There are several limitations that need to be considered when interpreting the results of this study. First, this was a small-sample, single-center study, and there is a selection bias, as there may be differences between regions. Second, although the reliability of the IES-R was good, it could not further differentiate the incidence of PTSD and evaluate overall PTSD symptoms. The results of this study may be different from those of other studies. Third, face-to-face follow-up interviews were performed in this study, subjects may have been impacted by the Hawthorne effect, and potential confounders were not adjusted, which may affect the results.

## 5. Conclusions

In summary, timely and effective multidisciplinary treatment may play a certain role in psychological relief, but psychological damage, which may last a long time, does not improve with the gradual improvement of quality of life or time. The psychological impacts of major public health events of worldwide concern cannot be ignored. Medical institutions and societal practices can control the degree of physical injury caused by COVID-19, but these institutions need to simultaneously address psychological damage early and provide continuous psychological support and treatment.

## Figures and Tables

**Figure 1 brainsci-12-00939-f001:**
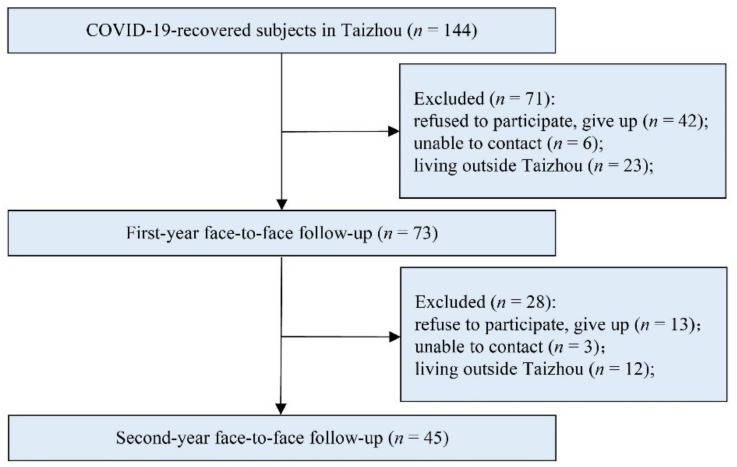
Screening flowchart.

**Figure 2 brainsci-12-00939-f002:**
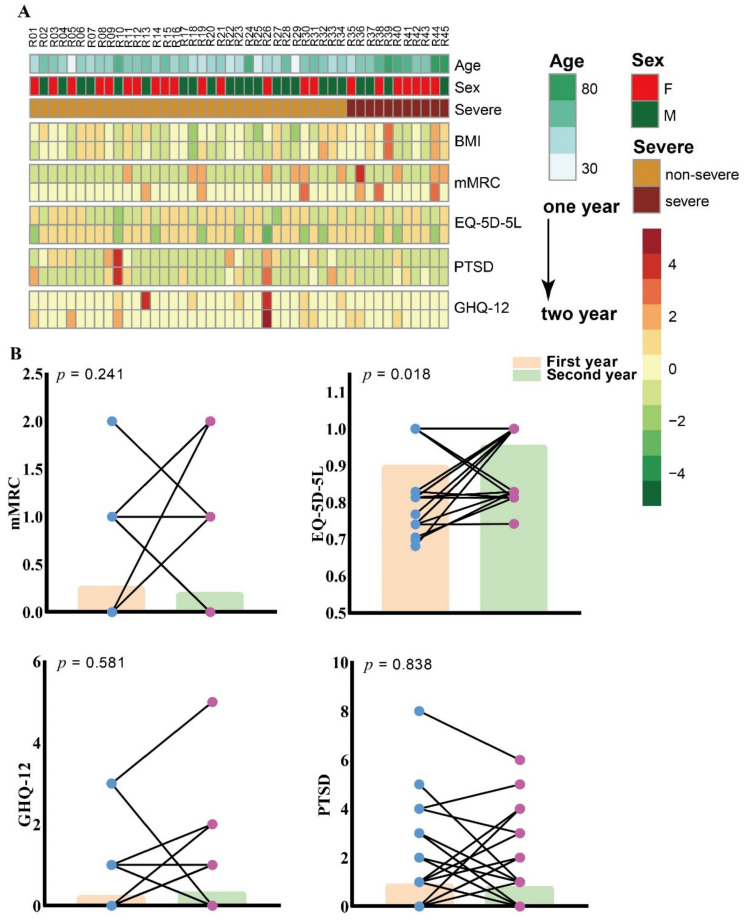
Changes in the mMRC, EQ-5D-5L, GHQ-12, and IES-R scores in the 45 recovered subjects. (**A**) Scores of each scale in two years for each subject. (**B**) The changes in each scale for recovered subjects.

**Figure 3 brainsci-12-00939-f003:**
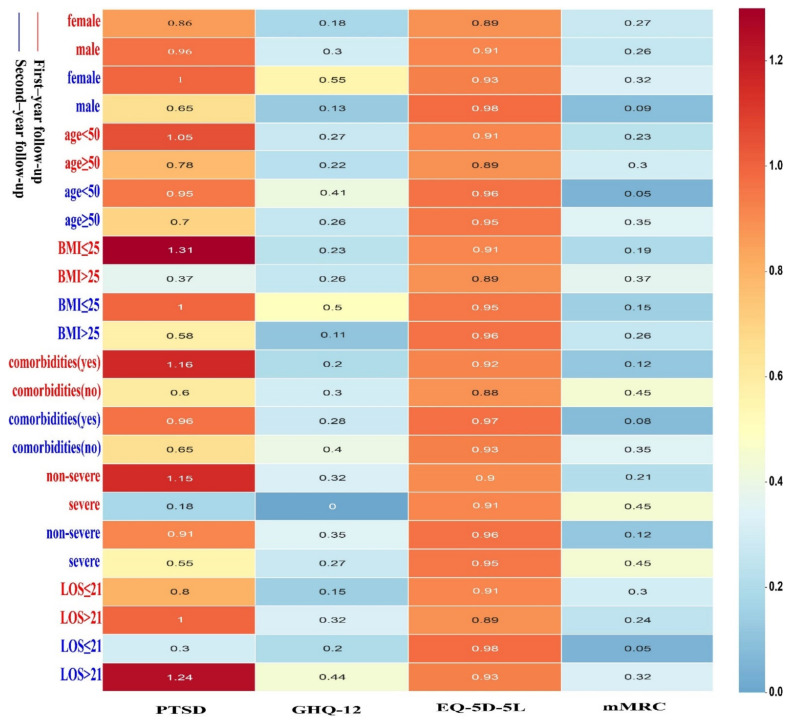
Changes in the scale scores of the 45 subjects in the first and second years classified by sex, age, BMI, comorbidities, disease severity, and LOS. Note: Mann-Whitney U test was used to analyze the differences of each scale in subgroups in the first and second year: female vs. male (0.93 vs. 0.98, *p* = 0.045) and LOS ≤ 21 vs. LOS > 21 (0.98 vs. 0.93, *p* = 0.034) in scales of EQ-5D-5L in the second year, comorbidities (yes) vs. comorbidities (no) (0.12 vs. 0.45, *p* = 0.029) in scales of mMRC in the first year, and BMI ≤ 25 vs. BMI > 25 (0.15 vs. 0.26, *p* = 0.025) in scales of mMRC in the second year.

**Table 1 brainsci-12-00939-t001:** Demographic characteristics of the subjects with who attended both follow-ups and only once follow-up.

	Overall Follow-Up (*n* = 73)		
	Both (*n* = 45)	Only Once (*n* = 28)	*p*-Value
Clinical characteristic			
Gender (Male/Female)	23/22	16/12	0.615
Median age (IQR), year	52 (43–57)	40 (37–49)	0.020
BMI (IQR), kg/m^2^	24.4 (22.5–26.5)	24.0 (22.1–26.8)	0.973
Severe *n* (%)	11 (24.4)	8 (28.6)	0.696
Comorbidities *n* (%)	20 (44.4)	6 (21.4)	0.046
LOS > 21 *n* (%)	25 (55.6)	11(39.3)	0.176

**Table 2 brainsci-12-00939-t002:** Demographic characteristics and mMRC, EQ-5D-5L, GHQ-12, and IES-R PTSD scores of recovered COVID-19-recovered subjects who completed both follow-ups.

	Both Follow-Up (*n* = 45)		
	First Year	Second Year	*p*
Clinical characteristic			
Gender (Male/Female)	23/22	23/22	-
Median age (IQR), year	51 (42–56)	52 (43–57)	0.553
BMI (IQR), kg/m^2^	24.4 (22.5–26.2)	24.4 (22.5–26.5)	0.881
Severe *n* (%)	11(24.0)	11(24.0)	-
Comorbidities *n* (%)	20 (44.0)	20 (44.0)	-
LOS > 21 *n* (%)	25 (55.6)	25 (55.6)	-
Scale			
mMRC	0.27 ± 0.50	0.20 ± 0.55	0.241
0 *n* (%)	43 (75.6)	39 (86.7)	-
≥1 *n* (%)	11 (24.4)	6 (13.3)	-
EQ-5D-5L	0.90 ± 0.12	0.95 ± 0.08	0.018
GHQ-12	0.24 ± 0.70	0.33 ± 0.88	0.581
≤2 *n* (%)	43 (95.6)	44 (97.8)	-
≥3 *n* (%)	2 (4.4)	1 ^a^ (2.2)	-
PTSD	0.91 ± 1.70	0.82 ± 1.51	0.838

The cutoff value of GHQ-12 is 3. ^a^ GhQ-12 score ≥ 3 in the first year. -: no comparison was made between the two groups.

**Table 3 brainsci-12-00939-t003:** Studies on mental health of COVID-19 survivors at 1-year follow-up.

First Author	Sample Size	Country	Mean/Median Age, Year	Male Proportion	Scale for Mental Health	Cutoff Value	Follow-Up Method	Mental Abnormal Proportion
Hidde Heesakkers [31]	246	The Netherlands	61.2	71.5%	Hospital Anxiety and Depression (HADS)	HADS-A ≥ 8 HADS-D ≥ 8	In-person visit or online	17.9% (44/246) with anxiety, 18.3% (45/246) with depression; mental symptoms were reported by 26.2% (64/244).
Yumiao Zhao [27]	94	China	48.1	57.4%	14-item Hamilton Anxiety Rating Scale (HAMA-14), the 24- item Hamilton Depression Rating Scale (HAMD-24),	HAMA ≥ 7 HAMD ≥ 7	In-person visit	41.5% (39/94) with anxiety, 42.6% (40/94) with depression.
Xin Huang [32]	537	China	56.2	51.9%	9-item Patient Health Questionnaire (PHQ-9), Generalized Anxiety Disorder Scale (GAD-7)	PHQ-9 ≥ 9 GAD-7 ≥ 9	In-person visit or online	6.3% (32/511) with anxiety, 11.9% (61/511) with depression.
Verena Rass [11]	81	Austria	54	59.3%	HADS	HADS-A ≥ 8 HADS-D ≥ 10	In-person visit	29% (24/81) with anxiety, 6% (5/81) with depression.
Martina Betschart [26]	43	Switzerland	60	69.8%	HADS	HADS-A ≥ 8 HADS-D ≥ 8	In-person visit	16% (6/38) with anxiety, 22% (8/36) with depression.
Roberta Ferrucci [33]	53	Italy	58.5	71.7%	Beck’s Depression Inventory-II (BDI-II)	BDI-II ≥ 14	In-person visit	26.4 (14/53) with depression.
Mario Gennaro Mazza [10]	192	Italy	59.2	68.2%	State-Trait Anxiety Inventory form Y (STAI-Y), Zung Severity Rating Scale (ZSDS)	STAI ≥ 40 ZSDS index ≥ 50	Online	40.1% (77/192) with anxiety, 30.7% (59/192) with depression.
Nicola Latronico [34]	51	Italy	NA	NA	HADS	HADS-A ≥ 8 HADS-D ≥ 8	In-person visit	17.8% (8/45) with anxiety, 9.8% (5/51) with depression.
Natalie Lorent [35]	222	Belgium	NA	NA	HADS	HADS-A ≥ 8 HADS-D ≥ 8	In-person visit	19% (40) with anxiety, 15% (32) with depression
Jennifer A. Frontera [36]	113	USA	64	65.5%	Quality of Life in Neurological Disorders (NeuroQoL)	Anxiety T-score > 60, Depression T-score > 60	In-person visit	8.5% (9/105) with anxiety, 6.7% (7/105) with depression.
Juan Pan	45	China	51	51.1%	GHQ-12	GHQ-12 ≥ 3	In-person	4.4% (2/45) with abnormal mental symptoms.

## Data Availability

The data presented in this study are available on request from the corresponding author.

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
