# Peer review of "Quality of Life and Mental Health Status in Recovered COVID-19 Subjects at Two Years after Infection in Taizhou, China: A Longitudinal Cohort Study"

_brainsci, 2022, doi:10.3390/brainsci12070939_

Round 1
Reviewer 1 Report
Summary: In the present manuscript, the authors performed a longitudinal study of long-term psychological consequences in recovered COVID-19 patients. They found that the only a very small fraction of the recovered subjects had any psychological problems, and these psychological problems did not improve with time, or at least for the duration for which these patients were tested. The study covers an important topic as psychological problems are affecting COVID-19 patients post recovery. The article is well-written and adequate details are provided for different section. I don’t have any major concern. Some minor concerns are given below:
Minor:
1) There is repetition of the same information including the values in the tables and text in several places. This can be consolidated. For example – section 3.2 and table-2.
2) Please rephrase line 153.
3) Please discuss section 3.3 and table 3 in the discussion and not results.
4) Please rephrase line 221.
Reviewer 2 Report
This is an interesting study dealing with a timely topic regarding mental health and quality of life of COVID-19 survivors. However, it has methodological flaws that make inferences from your results difficult to believe. The authors might re-arrange analyses.
Abstract
-You should provide statistical methods conducted in your study and estimates in accordance with a prospective cohort study (eg., hazard ratios). A prevalence percentage is not enough evidence.
Introduction
-Given the background provided in this section, would you state any hypothesis for your study? It seems as very plausible that COVID-19 patients might have some sort of sequelae after one- or two-years follow-up.
Methods
-Lines 76-78. How did you confirm that these were COVID-19 patients? How were they recruited? Were there recruited at convenience? If yes, then a selection bias is more than likely. Please, provide information on these critical points.
-Here it comes my main concern. You conducted any type of t-test between the two samples, but you don´t state if it was paired t-test or for independent groups. Importantly, because you did not use any regression model adjusting for potential confounders the results of your study are possibly wrong. Also, your sample, particularly in the second follow-up is small, and you might don´t have the enough statistical power to detect differences. I would have conducted a Cox-regression.
Results
-You might need to rewrite this part if you re-arrange your analyses.
Discussion
-You might need to also rewrite this part if you conduct different analyses.
-In the limitations, don´t forget to comment about the potential selection bias and the worryingly residual confounding threating your results.
Reviewer 3 Report
REVIEW
I congratulate the Authors for the work done.
In my opinion, the work called ‘Quality of life and mental health status in recovered COVID-19 subjects at two years after infection: a longitudinal cohort study’ is written very well, the methodology research part is well described, and the discussion is written in an interesting and appropriate way.
I have only three minor comments that could improve this manuscript:
1. The research was carried out on a relatively small group of respondents, therefore I suggest you include in the title information that these are preliminary reports.
2. The figures (2 and 3) in the results section are illegible. I suggest you improve them.
3. The references part has double numbering. Remove double numbering.
Round 2
Reviewer 2 Report
The authors reasonably adressed my queries.